# Prediction of Second-Order Rate Constants of Sulfate Radical with Aromatic Contaminants Using Quantitative Structure-Activity Relationship Model

**Han Ding** [ID] **and Jiangyong Hu** *

Department of Civil & Environmental Engineering, National University of Singapore, 1 Engineering Drive 2, Singapore 117576, Singapore; dingh09@u.nus.edu
* Correspondence: ceehujy@nus.edu.sg

**Abstract:** Predicting the second-order rate constants between aromatic contaminants and a sulfate radical ($k_{SO_4^{\bullet-}}$) is vital for the screening of pollutants resistant to sulfate radical-based advanced oxidation processes. In this study, a quantitative structure-activity relationship (QSAR) model was developed to predict the values for aromatic contaminants. The relationship between $\log k_{SO_4^{\bullet-}}$ and three molecular descriptors (electron density, steric energy, and ratio between oxygen atoms and carbon atoms) was built through multiple linear regression. The goodness-of-fit, robustness, and predictive ability of the model were characterized statistically with indicators showing that the model was reliable and applicable. Electron density was found to be the most influential descriptor that contributed the most to $\log k_{SO_4^{\bullet-}}$. All data points fell within the applicability domain, and no outliers existed in the training set. The comparison with other models indicates that the QSAR model performs well in elucidating the mechanism of the reaction between aromatic compounds and sulfate radicals.

**Keywords:** QSAR; rate constants; sulfate radical; aromatic compounds





## 1. Introduction

Sulfate radical-based advanced oxidation processes (SR-AOPs) are considered as a promising technology for the treatment of wastewater with recalcitrant organic contaminants [1]. With a high standard reduction potential comparable to that of HO$^\bullet$ ($E^\circ(SO_4^{\bullet-} / SO_4^{2-})$ = 2.60 V vs. $E^\circ(HO^\bullet/OH^-)$ = 2.80 V), the sulfate radical ($SO_4^{\bullet-}$) is highly reactive and capable of mineralizing recalcitrant organic contaminants [2]. Unlike HO$^\bullet$, which reacts unselectively with organic compounds via hydrogen abstraction or addition, $SO_4^{\bullet-}$ prefers to attack organics rich in electron moieties through the single-electron transfer reaction [3]. Therefore, $SO_4^{\bullet-}$ is less affected by the complex water matrix in real wastewaters than HO$^\bullet$. Moreover, $SO_4^{\bullet-}$ has a much longer lifetime than HO$^\bullet$, making its reaction with pollutants more efficient [4]. Peroxydisulfate and peroxymonosulfate have been frequently used as the precursors of $SO_4^{\bullet-}$, which are more stable and safer for transportation than the commonly used HO$^\bullet$ precursor $H_2O_2$ [5]. The advantages of $SO_4^{\bullet-}$ make SR-AOPs a potential substitute to HO$^\bullet$-based AOPs.

Aromatic compounds are widely produced and used. Some synthetic aromatic compounds such as pesticides and pharmaceuticals are resistant to conventional wastewater treatment processes and discharged directly into the aquatic environment [6]. They constitute a major group of organic pollutants in the aquatic ecosystem and pose threats to human health due to their potential carcinogenicity [7,8]. SR-AOPs have been extensively studied for the removal of recalcitrant aromatic contaminants [9]. The second-order rate constants of $SO_4^{\bullet-}$ with organic compounds ($k_{SO_4^{\bullet-}}$) in aqueous solution range from $10^5$ to $10^9$ M$^{-1}$ s$^{-1}$ [10]. Considering that the scavenging of $SO_4^{\bullet-}$ by inorganic anions and natural organic matters in the water matrix will inhibit the degradation of target contaminants [11],

only contaminants with higher $k_{SO_4^{\bullet-}}$ than those compounds in the water matrix could be removed efficiently in real waters. Currently, the available $k_{SO_4^{\bullet-}}$ values are quite limited and obtained mainly through experiments [12]. As there are thousands of aromatic contaminants, experimental measurement of all the $k_{SO_4^{\bullet-}}$ values would be laborious, costly, and time consuming. Therefore, it is necessary to develop an alternative method to estimate the $k_{SO_4^{\bullet-}}$ values as they are imperative to the assessment of whether a target compound is suitable for SR-AOPs treatment.

Quantitative structure-activity relationship (QSAR) analysis focuses on correlating the molecular descriptors calculated from molecular structures with the activities of the corresponding chemicals (e.g., toxicity and biodegradability) [13–15]. QSAR models are mainly used to predict a particular physical or chemical property of a chemical compound and interpret the mechanism behind the prediction [16]. Numerous QSAR models have been developed to predict the rate constants of reactive species such as $HO^{\bullet}$, $O_3$, singlet oxygen, and hydrated electron with organic contaminants [17–20]. However, QSAR models for $SO_4^{\bullet-}$ are quite limited. Xiao et al. [21] found that $k_{SO_4^{\bullet-}}$ was negatively related to the ratio of oxygen atoms to carbon atoms and the energy gap between the lowest occupied molecule orbital and highest occupied molecule orbital. Unfortunately, their model did not contain a variable positively correlated with the $k_{SO_4^{\bullet-}}$ values; thus, it is unable to tell what property of the chemical compound would facilitate its reaction with $SO_4^{\bullet-}$. Ye et al. [12] developed a model that linked $\ln k_{SO_4^{\bullet-}}$ with 32 frequencies of structural fragment. However, too many independent variables in the model may cause overfitting problems and reduce the predictive ability of the model [22]. Much more effort could be made to improve the QSAR models for $k_{SO_4^{\bullet-}}$ prediction due to a lack of relevant studies currently.

In this study, a QSAR model for predicting the $k_{SO_4^{\bullet-}}$ of aromatic compounds was developed and validated by using the multiple linear regression (MLR) method. Two new descriptors (electron density and steric energy) were incorporated into the model to bring new insight into the $SO_4^{\bullet-}$ reaction with aromatic compounds. Following the guidance of The Organization for Economic Cooperation and Development (OECD) for QSAR development [23], statistical characteristics of the developed model were analyzed, the mechanism behind the model was interpreted, and the applicability domain of the model was assessed. Finally, a comparison with previous models was made. The results of this work would help to judge whether an aromatic compound is suitable for SR-AOPs treatment.

## 2. Materials and Methods

### 2.1. Dataset

The $k_{SO_4^{\bullet-}}$ values ($M^{-1}s^{-1}$) of 88 aromatic compounds were collected from published literature, and the corresponding references are cited in Table S1. Five descriptors, i.e., electron density, steric energy (kcal/mol), the ratio between the number of oxygen atoms and carbon atoms, volume of the molecule ($Å^3$), and octanol–water partitioning coefficient were calculated to represent the physical properties of selected chemicals. The electron density ($E$) represents the probability of an electron appearing in a specific space around an atom or molecule. The electron density on each atom of an aromatic compound was calculated by Gaussian 16 Rev. A.03 with HF method at 6-31G level [24]. The highest electron density on the benzene ring was selected as the descriptor $E$. Steric energy ($S$) is the sum of energies that resulted from bonded and non-bonded energies within a molecule, which reflects the energy due to the geometry of a molecule [25]. The calculation of $S$ was run on the Chem3D 20.1.1 with the MM2 Dynamics method. The ratio between the number of oxygen atoms and carbon atoms (O/C) of a molecule was found to be negatively correlated with $k_{SO_4^{\bullet-}}$ [21]. The molecular volume ($V$) and octanol–water partition coefficient (logP) of each compound were obtained from Molinspiration Cheminformatics free web services (https://www.molinspiration.com, accessed date: 25 June 2021). The $\log k_{SO_4^{\bullet-}}$ and descriptor values of selected aromatic compounds are listed in Table S1. The

88 aromatic compounds were randomly divided into a training set and validation set with a ratio of 3:1. The data in the training set were used to develop the QSAR model, while the data in the validation set were used to verify the predictive ability of the developed model. MATLAB R2021b was used for the generation of the QSAR model and its validation.

## 2.2. QSAR Model Development and Characterization

The QSAR model in this study depicts the linear relationship between $\log k_{SO_4^{\bullet-}}$ and the molecular descriptors, as shown in Equation (1). A stepwise multiple linear regression method was used to determine the significant descriptors in the model. A model with all five descriptors was built first. Then, the *p*-value of each descriptor was calculated. The descriptors with a *p*-value lower than the significance level (0.05) were considered as insignificant and then excluded from the model one by one until all the descriptors left were significant.

$$\log k_{SO_4^{\bullet-}} = \beta_0 + \beta_1 E + \beta_2 S + \beta_3 (O/C) + \beta_4 \log V + \beta_5 \log P \tag{1}$$

The degree of multicollinearity among the descriptors was reflected by the variance inflation factor (VIF), which was calculated by Equation (2) [26]. The $r_i^2$ is the coefficient of determination when conducting multiple linear regression between the *i*th descriptor and all the other descriptors. A value of 10 was suggested as the threshold of VIF, above which the multicollinearity is considered as severe [27]. The goodness-of-fit of the model was assessed by $R^2$, adjusted $R^2 (R_{adj}^2)$ and the root mean square error (RMSE). $R_{adj}^2$ and RMSE were calculated by Equations (3) and (4), respectively, where $n$ is the number of compounds in the training set, $k$ is the number of descriptors, $y_i$ is the experimental $\log k_{SO_4^{\bullet-}}$ of the *i*th compound, and $\hat{y}_I$ is the predicted $\log k_{SO_4^{\bullet-}}$. Leave-one-out cross-validation was conducted to estimate the robustness of the developed model. For all compounds in the training set, one compound was removed from the training set each time to test the model, and the rest of the compounds were used to train the model [28]. The indicator of leave-one-out cross-validation $Q_{LOO}^2$ was calculated by Equation (5), where $\hat{y}_I$ is the $\log k_{SO_4^{\bullet-}}$ predicted from model with the *i*th compound removed from the training set, and $\overline{y_t}$ is the average experimental $\log k_{SO_4^{\bullet-}}$ of the training set [28]. $Q_{LOO}^2$ measures the robustness of the developed model [29].

$$\text{VIF} = \frac{1}{1 - r_i^2} \tag{2}$$

$$R_{adj}^2 = 1 - \left(\frac{n-1}{n-k-1}\right)\left(1 - R^2\right) \tag{3}$$

$$\text{RMSE} = \sqrt{\frac{\sum_{i=1}^{n}(y_i - \hat{y}_i)^2}{n}} \tag{4}$$

$$Q_{LOO}^2 = 1 - \frac{\sum_{i=1}^{n}(y_i - \hat{y}_i)^2}{\sum_{i=1}^{n}(y_i - \overline{y_t})^2} \tag{5}$$

## 2.3. Validation of the Model

The validation set was used to test the predictive ability of the model developed from the training set. The indicator $Q_{ext}^2$ was defined as Equation (6), where $y_j$ is the experimental $\log k_{SO_4^{\bullet-}}$ of the *j*th compound in the validation set, $\hat{y}_j$ is the predicted $\log k_{SO_4^{\bullet-}}$ of the *j*th compound in the validation set, and $\overline{y_t}$ is the average $\log k_{SO_4^{\bullet-}}$ of all compounds in the training set. A higher $Q_{ext}^2$ value means a better prediction by the model, and 0.5 was

suggested as the threshold [30,31]. The external RMSE was calculated by Equation (7), which gauges the deviation of predicted values from the experimental ones.

$$Q^2_{\text{ext}} = 1 - \frac{\sum_{j=1}^{test} (y_j - \hat{y}_j)^2}{\sum_{j=1}^{test} (y_j - \overline{y_t})^2} \tag{6}$$

$$\text{RMSE}_{\text{ext}} = \sqrt{\frac{\sum_{j=1}^{test} (y_j - \hat{y}_j)^2}{n}} \tag{7}$$

Moreover, the conditions below should be met so that the predictive ability of the model could be considered satisfactory [30]:

$$R^2_{\text{ext}} > 0.6 \tag{8}$$

$$\frac{R^2_{\text{ext}} - R^2_0}{R^2_{\text{ext}}} < 0.1 \text{ or } \frac{R^2_{\text{ext}} - R'^2_0}{R^2_{\text{ext}}} < 0.1 \tag{9}$$

$$0.85 \leq k \text{ and } k' \leq 1.15 \tag{10}$$

where $R_{\text{ext}}$ is the correlation coefficient between the predicted and experimental $\log k_{\text{SO}_4^{\bullet-}}$ values; $R^2_0$ is the coefficient of determination when the experimental $\log k_{\text{SO}_4^{\bullet-}}$ is regressed against the predicted $\log k_{\text{SO}_4^{\bullet-}}$ with the fitting curve through the origin and $k$ is the corresponding slope; $R'^2_0$ is the coefficient of determination when the predicted $\log k_{\text{SO}_4^{\bullet-}}$ is regressed against the experimental $\log k_{\text{SO}_4^{\bullet-}}$ with the fitting curve through the origin, and $k'$ is the corresponding slope.

A y-randomization test was used to verify the robustness of the developed model by measuring the degree of chance correlation between $\log k_{\text{SO}_4^{\bullet-}}$ and the descriptors [32]: the randomly shuffled $\log k_{\text{SO}_4^{\bullet-}}$ values in the training dataset were regressed against the fixed descriptors for several times to generate new MLR models, of which the $R^2$ shall be poor. The y-randomization test was conducted with Scikit-learn 1.0.2 [33] and the code was used to calculate the $R^2$ for each shuffle was listed in Text S1.

*2.4. Relative Contribution of Each Descriptor*

The relative contribution of each descriptor to the predicted $\log k_{\text{SO}_4^{\bullet-}}$ was estimated with Equation (11), where $E_i$, $S_i$, and $(O/C)_i$ are the descriptors of the $i$th compound, and $D_i$ is one of the three descriptors in the model. The analysis of relative contribution would help to elaborate on the role of each descriptor in the model.

$$\text{Relative contribution (\%)} = \frac{|\beta_i D_i|}{\beta_0 + |\beta_1 E_i| + |\beta_2 S_i| + |\beta_3 (O/C)_i|} \times 100 \tag{11}$$

*2.5. Applicability Domain*

The applicability domain (AD) is used to define a region of chemicals with specific structures where the model could make an accurate prediction [29]. The visualization of AD was fulfilled by the Williams plot, of which the $X$-axis refers to hat values (leverages), and the $Y$-axis refers to standardized residuals ($\delta$) [34]. The hat value was calculated with Equation (12), where $h_i$ is the hat value of the $i$th compound, $X$ is the $n \times k$ descriptor matrix containing all compounds in the training set, and $x_i$ is the $1 \times k$ descriptor vector of the $i$th compound ($n$ is the number of compounds in the training set and $k$ is the number of descriptors) [35]. The critical hat value ($h^*$) was calculated with Equation (13) [36]. When $h_i < h^*$ and $\delta < 3$, the predicted $\log k_{\text{SO}_4^{\bullet-}}$ of the $i$th compound is considered reliable [21]. The standardized residual of the $i$th compound was calculated with Equation (14) [37].

$$h_i = x_i (X^T X)^{-1} x_i^T \tag{12}$$

$$h^* = \frac{3(k+1)}{n} \qquad (13)$$

$$\delta_i = \frac{y_i - \hat{y}_i}{\sqrt{\frac{\sum_{i=1}^{n}(y_i - \hat{y}_i)^2}{n-k-1}}} \qquad (14)$$

## 3. Results

### 3.1. Selection of Significant Descriptors

Backward stepwise regression was performed on the training set ($n$ = 66) to select significant descriptors for the model. As shown in Table 1, when all five descriptors were included in the model, three descriptors (O/C, logV, and logP) were found to be insignificant, as their $p$-values were above the significance level of 0.05.

**Table 1.** Selection of significant descriptors for the QSAR model by backward stepwise regression ($n$ = 66).

| Steps | Descriptors | Coefficient | $t$-Statistic | $p$-Value | $R^2$ | $R^2_{\text{adj}}$ | RMSE | Decision |
|---|---|---|---|---|---|---|---|---|
| 1 | $E$ | 1.5755 | 5.7113 | 0.0000 | 0.685 | 0.659 | 0.219 | Exclude logP |
| | $S$ | 0.0042 | 6.5814 | 0.0000 | | | | |
| | O/C | −0.1924 | −0.5424 | 0.5895 | | | | |
| | Log$V$ | −0.2572 | −1.2209 | 0.2269 | | | | |
| | Log$P$ | 0.0374 | 1.4067 | 0.1647 | | | | |
| | Constant | | 1.8972 | | | | | |
| 2 | $E$ | 1.6799 | 6.2729 | 0.0000 | 0.675 | 0.653 | 0.221 | Exclude log$V$ |
| | $S$ | 0.0039 | 6.4515 | 0.0000 | | | | |
| | O/C | −0.5938 | −2.7933 | 0.0070 | | | | |
| | Log$V$ | −0.1412 | −0.7225 | 0.4727 | | | | |
| | Constant | | 1.2381 | | | | | |
| 3 | $E$ | 1.6882 | 6.3344 | 0.0000 | 0.672 | 0.656 | 0.220 | Accept $E$, $S$ and O/C as the significant descriptors |
| | $S$ | 0.0037 | 6.7496 | 0.0000 | | | | |
| | O/C | −0.6035 | −2.8559 | 0.0058 | | | | |
| | Constant | | 0.8868 | | | | | |

After excluding logP from the model, O/C became significant while logV was still insignificant. The decrease in $R^2$ and $R^2_{\text{adj}}$ and the increase in RMSE were results from reducing the overfitting by taking out logP. Finally, the logV was removed from the model, and the descriptors left were all significant. Meanwhile, $R^2_{\text{adj}}$ was increased and RMSE was decreased, indicating that the overfitting was reduced with an unnecessary descriptor removed. The formula of the model is shown below:

$$\log k_{\text{SO}_4^{\bullet-}} = 0.8868 + 1.6882E + 0.0037S - 0.6035(\text{O/C})$$
$$n = 66, \quad R^2 = 0.672, \quad R^2_{\text{adj}} = 0.656, \quad \text{RMSE} = 0.220, \quad F = 42.30, \quad p < 0.0001 \qquad (15)$$

### 3.2. Exclusion of Outliers

The analysis of residuals of the training set could help to identify outliers that affected the goodness-of-fit of the model. As shown in Figure 1, the compounds whose residual confidence intervals did not include zero point were classified as outliers and colored in red. After removing the identified outliers, the MLR would be repeatedly conducted to find new outliers until all the residual confidence intervals included the zero point.

After four rounds of MLR, five compounds in the training set were removed to improve the goodness-of-fit of the model. The final formula of the developed model is shown in Equation (16). By removing five outliers, $R^2$ was increased from 0.672 to 0.748, and RMSE was decreased from 0.220 to 0.193, indicating that the goodness-of-fit was significantly improved.

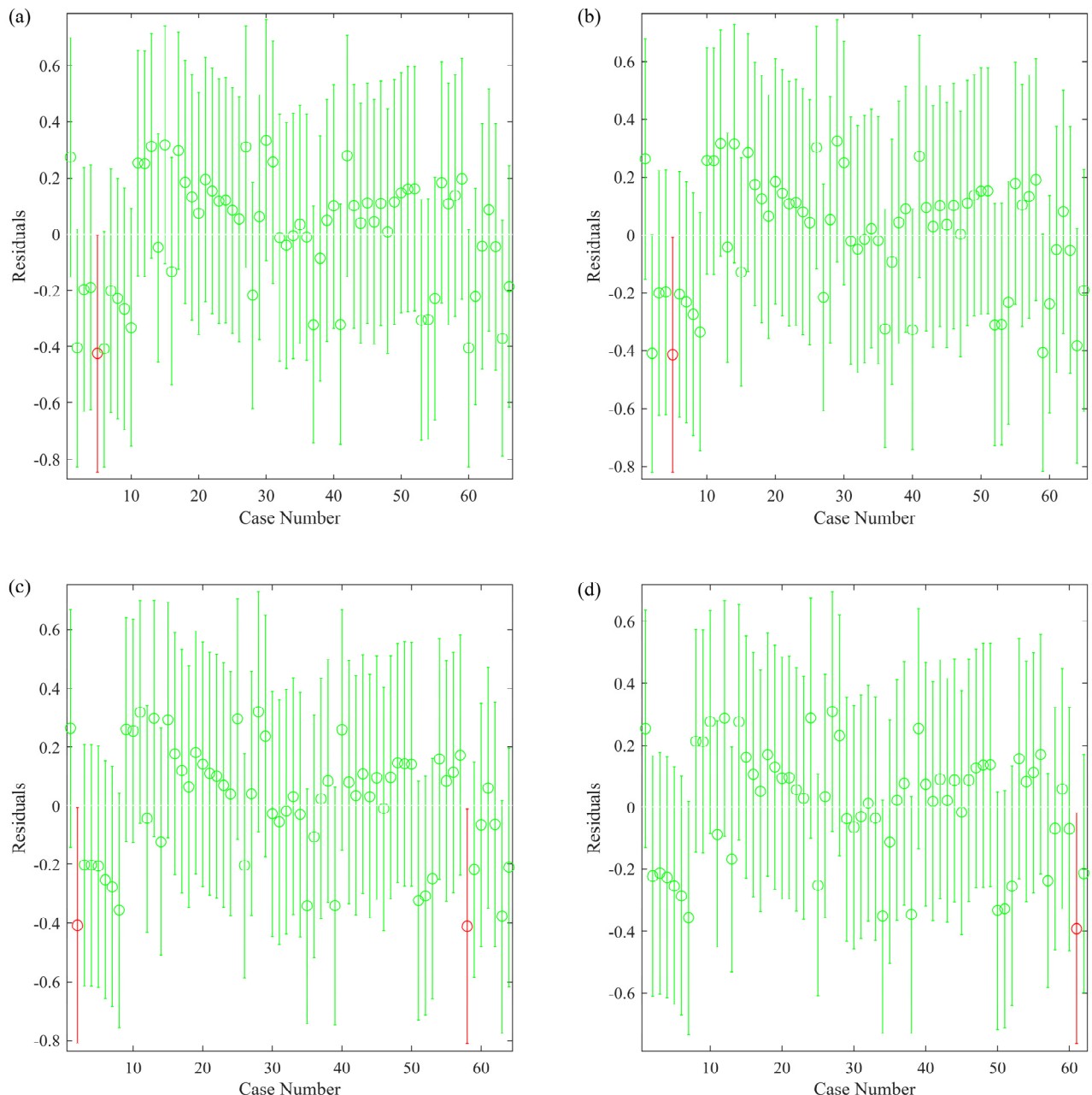

**Figure 1.** Identification of outliers in the training set by residual analysis: (**a**) first round; (**b**) second round; (**c**) third round; (**d**) fourth round. Red circles represent the outliers identified in each round.

$$\log k_{SO_4^{\bullet-}} = 0.4641 + 1.7751E + 0.0036S - 0.5836(O/C)$$

$$R^2 = 0.748, \ R_{adj}^2 = 0.735, \ \text{RMSE} = 0.193, \ F = 56.43, \ p < 0.0001 \quad (16)$$

### 3.3. Statistical Characteristics of the Developed Model

The statistical indicators of the model before and after excluding the five outliers are listed in Table 2. After excluding the outliers from the training set, the goodness-of-fit and robustness of the model were both improved. Even though $Q_{ext}^2$ and $\text{RMSE}_{ext}$ were slightly decreased, $R_{ext}^2$ was increased. Therefore, the predictive ability of the model remained stable after the exclusion of outliers.

**Table 2.** Statistical characterization of the model before and after excluding the outliers.

| Model | Goodness-of-Fit | | | Robustness | Predictive Ability | | | $R_0^2$ | $R_0'^2$ | $k$ | $k'$ | $\frac{R_{ext}^2 - R_0^2}{R_{ext}^2}$ | $\frac{R_{ext}^2 - R_0'^2}{R_{ext}^2}$ |
| --- | --- | --- | --- | --- | --- | --- | --- | --- | --- | --- | --- | --- | --- |
| | $R^2$ | $R_{adj}^2$ | RMSE | $Q_{LOO}^2$ | $Q_{ext}^2$ | $RMSE_{ext}$ | $R_{ext}^2$ | | | | | | |
| Including outliers | 0.672 | 0.656 | 0.220 | 0.617 | 0.605 | 0.282 | 0.632 | 0.066 | 0.610 | 1.010 | 0.990 | 0.896 | 0.035 |
| Excluding outliers | 0.748 | 0.735 | 0.193 | 0.694 | 0.603 | 0.289 | 0.648 | 0.110 | 0.624 | 1.013 | 0.987 | 0.830 | 0.037 |

The conditions listed in inequalities 8–10 were all met, showing a satisfactory predictive ability of the model. The VIF of *E*, *S*, and O/C were 1.48, 1.11, and 1.32, respectively. Therefore, the multicollinearity among the variables was negligible, as the VIF values were well below the threshold of 10 and close to 1. As shown in Figure 2, the values of descriptors were far away from the flat surfaces, which represent the fitting results of *E*, *S*, and O/C by MLR when the other two descriptors acted as independent variables. It confirms that there is little multicollinearity among *E*, *S*, and O/C, suggesting the model has high stability [38].

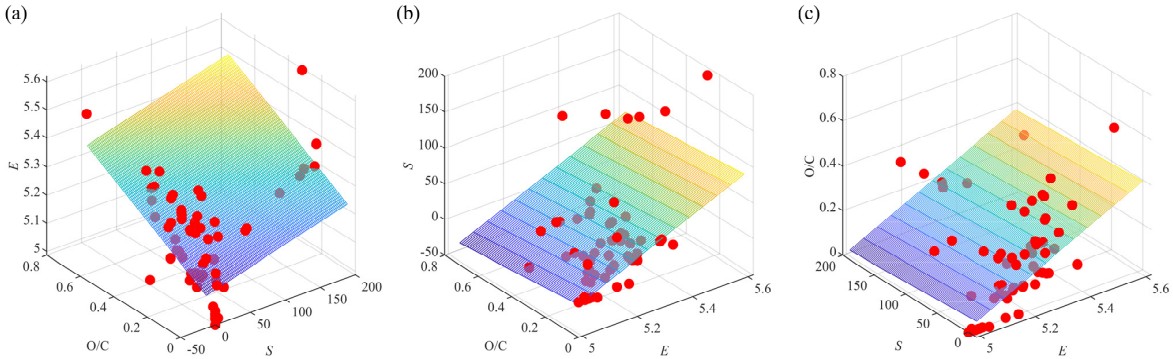

**Figure 2.** Multicollinearity analysis among descriptors: (**a**) *E*; (**b**) *S*; (**c**) O/C.

The predicted $\log k_{SO_4^{\bullet-}}$ versus the experimental $\log k_{SO_4^{\bullet-}}$ is shown in Figure 3. The prediction made by the QSAR model agreed quite well with the experimental results, showing a high predictive ability of the developed model. The result of the y-randomization test is shown in Figure S1. The poor $R^2$ values (<0.16) for the MLR models developed from shuffled $\log k_{SO_4^{\bullet-}}$ against *E*, *S*, and O/C indicate that there is no chance correlation. Therefore, the MLR model developed (Equation (16)) can be trusted to predict $\log k_{SO_4^{\bullet-}}$ values from new descriptors.

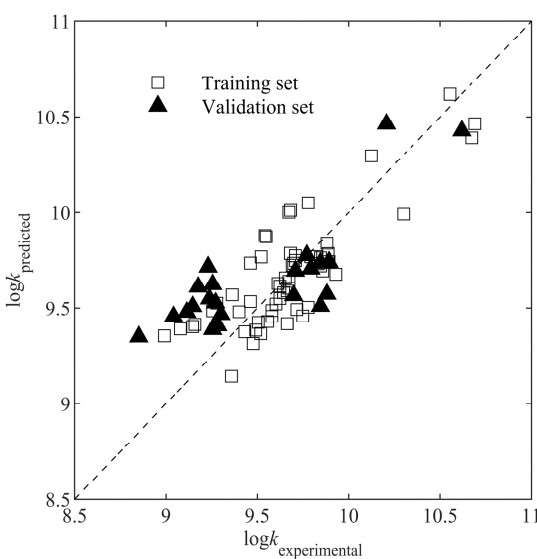

**Figure 3.** Predicted $\log k_{SO_4^{\bullet-}}$ versus experimental $\log k_{SO_4^{\bullet-}}$.

## 4. Discussion

### 4.1. Interpretation of the Model

The coefficient of $E$ is 1.7751, as shown in Equation (16), which is much larger than the other two coefficients, indicating that the change of $\log k_{SO_4^{\bullet-}}$ per unit $E$ is the highest. Therefore, $E$ is the most influential factor in the model. As shown in Figure 4, $E$ also contributed the most to $\log k_{SO_4^{\bullet-}}$. $SO_4^{\bullet-}$ reacts with aromatic compounds mainly through single electron transfer (SET) from the benzene ring to the radical. As the occurrence of the SET process requires the electron extraction from the nucleophile [39], the electron density of the benzene ring may play an essential role in the electron transfer rate between $SO_4^{\bullet-}$ and aromatic compounds. It was assumed in this study that the highest electron density on the carbon atom of the benzene ring was positively correlated with $\log k_{SO_4^{\bullet-}}$; i.e., a higher electron density would lead to a faster reaction rate, which is consistent with the result of the model, as the coefficient of $E$ is positive.

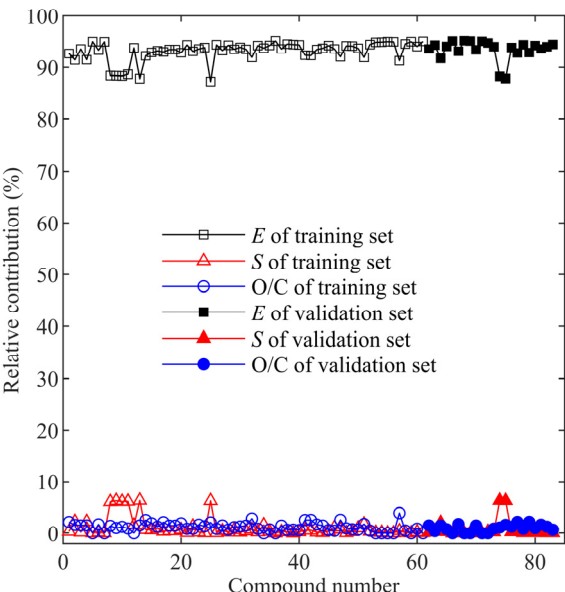

**Figure 4.** The relative contribution of three descriptors to $\log k_{SO_4^{\bullet-}}$.

As shown in Figure 5a, $R^2$ of $\log k_{SO_4^{\bullet-}}$ vs. $E$ is 0.455, suggesting a moderate correlation between the rate constant and electron density. However, Luo et al. [40] observed no clear relationship between the Gibbs free energy of SET ($\Delta G^{\circ}_{SET}$, which represents the reactivity of $SO_4^{\bullet-}$ toward aromatic contaminants) and the total electrostatic potential charge on the benzene ring (ESP, which represents the electron density on the benzene ring). The disparity might be due to the different ways in calculating electron density. In this study, the highest rather than the total electron density on the carbon atom in the benzene ring was used as it was assumed that $SO_4^{\bullet-}$ might preferentially attack the carbon atom in the benzene ring with the highest electron density.

The coefficient of $S$ is 0.0036, of which the absolute value is the least among the three coefficients. Therefore, $S$ is the least influential factor in the model. As shown in Figure 4, the relative contribution of $S$ was comparable to that of O/C for most of the aromatic compounds selected, except for those with the benzenesulfonamide moiety, which shows a much higher contribution of $S$ than that of O/C. The steric energy is a sum of energies from bond stretching, bending, torsion, Van der Waals, and electrostatic interactions within a molecule [41]. The lowest energy conformation of a molecule is most favored, and it is achieved when the steric energy is minimized [25]. As the reactivity of a molecule is affected by its geometry, there should be a certain kind of relationship between the steric energy and the reactivity of a molecule. The positive correlation between $S$ and $\log k_{SO_4^{\bullet-}}$ indicates that higher steric energy tends to make the aromatic molecule more reactive

to $SO_4^{\bullet-}$. As shown in Figure 5b, $R^2$ of the linear regression between $S$ and $\log k_{SO_4^{\bullet-}}$ is 0.503, which is even higher than that between $E$ and $\log k_{SO_4^{\bullet-}}$. However, the data points in Figure 5b were closely gathered, while in Figure 5a, the data points were scattered around the regression line. Therefore, a higher $R^2$ of $\log k_{SO_4^{\bullet-}}$ vs. $S$ does not mean a higher correlation between $\log k_{SO_4^{\bullet-}}$ and $S$.

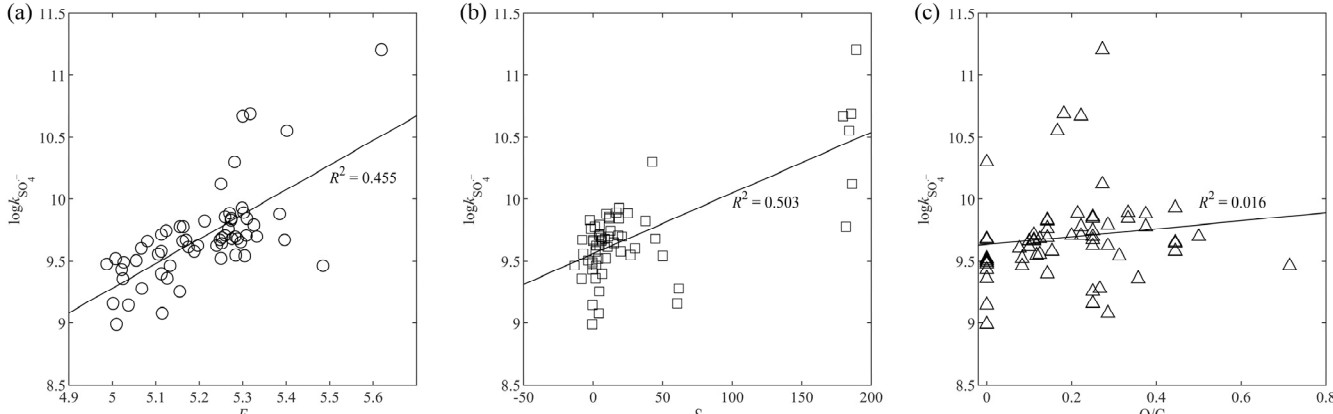

**Figure 5.** Correlation between $\log k_{SO_4^{\bullet-}}$ and each descriptor: (**a**) $E$; (**b**) $S$; (**c**) O/C.

The negative coefficient of O/C (−0.5836) indicates that the increase in oxygen content would inhibit the reactivity of aromatic compounds toward $SO_4^{\bullet-}$. Oxygen atoms, especially those attached directly to the benzene ring, have a strong ability to withdraw electrons from the benzene ring. As a result, the electron density on the benzene ring is reduced, and the electron transfer process is hindered. Xiao et al. [21] also reported that $\log k_{SO_4^{\bullet-}}$ was negatively correlated with O/C, and the correlation was quite strong as $R^2$ of $\log k_{SO_4^{\bullet-}}$ vs. O/C was 0.621. They suggested that an increase in O/C would decrease the number of H atoms, thus weakening the H abstraction by $SO_4^{\bullet-}$, which would slow down the reaction. However, no clear linear relationship was observed between $\log k_{SO_4^{\bullet-}}$ and O/C in this study (Figure 5c). There were only six compounds containing no oxygen atom in the training set (65 compounds in total) of Xiao et al. [21], while there were 11 in the training set of this study (61 compounds in total). The difference in training set composition might lead to the disparity in the importance of O/C to the models.

### 4.2. Applicability Domain

The standardized residual depicts the difference between the experimental and predicted results. As shown in Figure 6, all the δ values were within ±3, indicating no outlier existed. The $h_i$ value represents how far the $x_i$ value of the $i$th compound deviates from the average X value of all the compounds [42]. There was only one compound (gallate ion) in the training set with an $h_i$ larger than h*, which could be called a "good high leverage point", as it made the model more stable and accurate [42]. It was suggested that such a compound would have excessive influence during the model development process [34]. The developed model exhibits good extrapolating ability, as all data points from the validation set were within the AD, suggesting that the $\log k_{SO_4^{\bullet-}}$ of chemicals with structures similar to those in the training set may be reliably predicted.

### 4.3. Comparison with Other Models

Studies regarding the QSAR model for $k_{SO_4^{\bullet-}}$ prediction are quite limited, and only two relevant literatures could be found. The comparison between previous models and the one developed in this study is listed in Table 3.

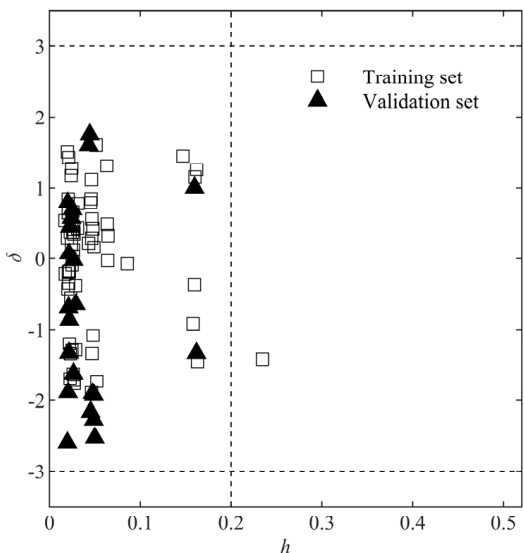

**Figure 6.** Williams plot for the QSAR model. The vertical dashed line is the boundary where *h* equals *h\**. The horizontal dashed lines are the boundaries where δ = ±3.

**Table 3.** A comparison of QSAR models for $k_{\mathrm{SO_4^{\bullet-}}}$ prediction.

| Reference | Model Type | *n* * | Molecular Descriptors | Goodness-of-Fit | | | Robustness | | Predictive Ability | | AD |
|---|---|---|---|---|---|---|---|---|---|---|---|
| | | | | $R^2$ | $R^2_{\mathrm{adj}}$ | RMSE | $Q^2_{\mathrm{LOO}}$ | $Q^2_{\mathrm{ext}}$ | $\mathrm{RMSE_{ext}}$ | $R^2_{\mathrm{ext}}$ | |
| Xiao et al. (2015) | MLR | 65 | Ratio of oxygen atoms to carbon; $E_{\mathrm{LUMO}}$ and $E_{\mathrm{HOMO}}$ energy gap | 0.866 | - | - | 0.86 | 0.89 | - | 0.89 | All but one compound from the validation set was outside the AD |
| Ye et al. (2017) | MLR and ANN | 75 | 32 molecular fragment descriptors | 0.88 (MLR); 0.99 (ANN) | - | - | - | - | - | 0.62 (MLR); 0.42 (ANN) | - |
| This study | MLR | 61 | Electron density, steric energy, and ratio of oxygen atoms to carbon | 0.748 | 0.735 | 0.193 | 0.694 | 0.603 | 0.289 | 0.648 | All data points of the validation set fell within the AD |

* The number of compounds in the training set.

Xiao et al. [21] developed the first QSAR model for $k_{\mathrm{SO_4^{\bullet-}}}$ prediction using the MLR method. Their model has a better goodness-of-fit, robustness, and predictive ability than the one in this study. However, the ratio of oxygen atoms to carbon atoms was the dominant descriptor in their model, which would limit the prediction of $k_{\mathrm{SO_4^{\bullet-}}}$ for compounds without oxygen atoms. Moreover, both descriptors in the model have negative coefficients, which means that the model could only reveal the factors that reduce the reactivity of compounds toward $\mathrm{SO_4^{\bullet-}}$. In this study, the model consists of two descriptors with positive coefficients and one descriptor with a negative coefficient. Therefore, both enhancing and inhibitory factors to the reaction between aromatic compounds and $\mathrm{SO_4^{\bullet-}}$ are explained by the model. Ye et al. [12] developed two models with 32 molecular fragment descriptors by using MLR and artificial neural network (ANN), respectively. They found that the ANN model showed much better goodness-of-fit but much lower predictive ability than MLR. Considering that the number of descriptors was close to the number of compounds in the training set, overfitting would weaken the predictive ability of the model developed by Ye et al. [12], which was also reflected by the low $R^2_{\mathrm{ext}}$ (0.62 for MLR and 0.42 for ANN). In addition, the descriptors in their model stand for the frequencies of structural fragments, which possess no physical meaning. Therefore, the mechanism behind the model was difficult to explain. In contrast, our model has successfully reduced the overfitting problem by excluding insignificant descriptors, and the physical properties of each descriptor help to elucidate the mechanism behind the reaction between aromatic compounds and $\mathrm{SO_4^{\bullet-}}$.

## 5. Conclusions

In this study, a QSAR model was developed and validated to predict the second-order rate constants between aromatic compounds and $SO_4^{\bullet-}$. The stepwise MLR was used to exclude insignificant descriptors, and the final model was composed of *E* (electron density), *S* (steric energy), and O/C (number of oxygen atoms vs. carbon atoms). Residual analysis was applied to remove outliers from the training set, and the goodness-of-fit of the model was improved. The statistical indicators for goodness-of-fit ($R_{adj}^2$ = 0.735), robustness ($Q_{LOO}^2$ = 0.694), and predictive ability ($Q_{ext}^2$ = 0.603 and $R_{ext}^2$ = 0.648) suggest that the model is satisfactory and applicable. *E* was the most influential descriptor and contributed most to the $\log k_{SO_4^{\bullet-}}$. The positive coefficient of *E* suggests that higher electron density on the benzene ring could enhance its reactivity to $SO_4^{\bullet-}$, which is consistent with the assumption that the reaction between $SO_4^{\bullet-}$ and aromatic compounds was achieved mainly through single electron transfer from the benzene ring to the radical. *S* is positively correlated with $\log k_{SO_4^{\bullet-}}$, suggesting that higher steric energy could improve the reactivity of aromatic compounds via affecting its geometry. An increase in O/C would decrease $\log k_{SO_4^{\bullet-}}$ as oxygen is electron withdrawing, so that the electron density on the benzene ring could be depleted. All data points in the validation set fell into the applicability domain, suggesting that the model is suitable for aromatic compounds with various functional groups. The comparison with other models shows that the QSAR model developed performed better in mechanism elucidation and overfitting reduction.

**Supplementary Materials:** The following supporting information can be downloaded at: https://www.mdpi.com/article/10.3390/w14050766/s1, Table S1: Values of molecular descriptors and $\log k_{SO_4^{\bullet-}}$ for selected aromatic compounds [43–75]; Text S1. Python code for the y-randomization test; Figure S1. The $R^2$ of shuffled $\log k_{SO_4^{\bullet-}}$ regressed against fixed *E*, *S* and O/C.

**Author Contributions:** Conceptualization, H.D., J.H.; methodology, H.D., J.H.; investigation, H.D.; writing—original draft preparation, H.D.; writing—review and editing, J.H.; visualization, H.D.; supervision, J.H.; project administration, J.H.; funding acquisition, J.H. All authors have read and agreed to the published version of the manuscript.

**Funding:** This research received no external funding.

**Institutional Review Board Statement:** Not applicable.

**Informed Consent Statement:** Not applicable.

**Data Availability Statement:** The data presented in this study is available on request from the corresponding author.

**Acknowledgments:** This research was supported by the President's Graduate Fellowship (GR-SUP0000003 President Grad Fel-PVO(SP)-IS) awarded by the National University of Singapore.

**Conflicts of Interest:** The authors declare no conflict of interest.

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
