# Peer review of "Prediction of Second-Order Rate Constants of Sulfate Radical with Aromatic Contaminants Using Quantitative Structure-Activity Relationship Model"

_water, doi:10.3390/w14050766_

Round 1

Reviewer 1 Report

This manuscript develops a quantitative structure-activity relationship model for predicting the second-order rate constants of sulfate radical with aromatic contaminants. The model shows better performance, such as high reliability and practicality than other counterparts. A new finding of electron density as the most influential descriptor for log?SO4•− determination is observed. The manuscript meets the high merit of publication in this journal. One suggestion is to improve the quality of the figures. Another comment is references should be cited in the discussion of "The ????•− values of 88 aromatic compounds were collected from published literature" (Line 80-81).

Author Response

pls see the attached file

Reviewer 2 Report

The authors were obtained QSAR model for second-order rate constants of sulfate radical for aromatic compounds. According to the below explanation, the final models could not interpret as predictive. So, the title should be changed to

 „Estimation of second-order rate constants of sulfate radical with  aromatic contaminants using quantitative structure-activity relationship model“

Validation of model:

  1. The model was not verified by Y-scrambling or/neither
  2. Value of parameters of the model with and without outliers (?????−???)/??? are not < 0.1. It means that models are not predictive. (Golbraikh and Tropsha, Journal of Computer-Aided Molecular Design, 16: 357–369, 2002)
  3. The models were not verified by Y-scrambling.
  4. The outliers should be explained.

The minor objections:

  1. The authors did not mention software for the generation of QSAR model and its validation.
  2. Line 80: Units of ????− is missing, as well as in Table 1.
  3. Did the authors perform backward stepwise regression on a full set of molecules or only on training? The number of compounds included in the calculation should be indicated in chapter 3.1. and Table 1, and under equation (17).
  4. Line 157: This is backward stepwise regression.
  5. CAS numbers in Table S1 are missing
  6. Figure 2 has low resolution. Maybe one graph of high quality is enough.

Author Response

pls see the attached file.

Round 2

Reviewer 2 Report

The authors answered and corrected the manuscript satisfactorily.